# Meta-Analysis illustrates possible role of lipopolysaccharide (LPS)-induced tissue injury in nasopharyngeal carcinoma (NPC) pathogenesis

David Z. Allen[ORCID][1]*, Jihad Aljabban[ORCID][2], Dustin Silverman[3], Sean McDermott[1], Ross A. Wanner[1], Michael Rohr[4], Dexter Hadley[5], Maryam Panahiazar[6]

1 The Ohio State College of Medicine, Columbus, Ohio, United States of America, 2 Department of Medicine, University of Wisconsin Hospital and Clinics, Madison, Wisconsin, United States of America, 3 Department of Otolaryngology, The Ohio State Wexner Medical Center, Columbus, Ohio, United States of America, 4 University of Central Florida, Orlando, Florida, United States of America, 5 Department of Pathology, University of Central Florida, Orlando, Florida, United States of America, 6 Department of Surgery, University of California San Francisco, San Francisco, California, United States of America

* david.allen@uth.tmc.edu

**Data Availability Statement:** We used samples from the GSE12452 [74], GSE13597[75], GSE53819[76], GSE40290[77], and GSE64634

## Abstract

### Background

Nasopharyngeal carcinoma (NPC) is a cancer of epithelial origin with a high incidence in certain populations. While NPC has a high remission rate with concomitant chemoradiation, recurrences are frequent, and the downstream morbidity of treatment is significant. Thus, it is imperative to find alternative therapies.

### Methods

We employed a Search Tag Analyze Resource (STARGEO) platform to conduct a meta-analysis using the National Center for Biotechnology's (NCBI) Gene Expression Omnibus (GEO) to define NPC pathogenesis. We identified 111 tumor samples and 43 healthy nasopharyngeal epithelium samples from NPC public patient data. We analyzed associated signatures in Ingenuity Pathway Analysis (IPA), restricting genes that showed statistical significance (p<0.05) and an absolute experimental log ratio greater than 0.15 between disease and control samples.

### Results

Our meta-analysis identified activation of lipopolysaccharide (LPS)-induced tissue injury in NPC tissue. Additionally, interleukin-1 (IL-1) and SB203580 were the top upstream regulators. Tumorigenesis-related genes such as homeobox A10 (HOXA10) and prostaglandin-endoperoxide synthase 2 (PTGS2 or COX-2) as well as those associated with extracellular matrix degradation, such as matrix metalloproteinases 1 and 3 (MMP-1, MMP-3) were also

[78] series. More information on the samples tagged within the studies and the raw data can be found at http://stargeo.org/analysis/612/.

**Funding:** The author(s) received no specific funding for this work.

**Competing interests:** The authors have declared that no competing interests exist.

upregulated. Decreased expression of genes that encode proteins associated with maintaining healthy nasal respiratory epithelium structural integrity, including sentan-cilia apical structure protein (SNTN) and lactotransferrin (LTF) was documented. Importantly, we found that etanercept inhibits targets upregulated in NPC and LPS induction, such as MMP-1, PTGS2, and possibly MMP-3.

## Conclusions

Our analysis illustrates that nasal epithelial barrier dysregulation and maladaptive immune responses are key components of NPC pathogenesis along with LPS-induced tissue damage.

## Introduction

Since the first published case in 1901, nasopharyngeal carcinoma (NPC) has been a well-known entity in otorhinolaryngology [1, 2]. Of epithelial origin, NPC is most commonly found in the fossa of Rosenmüller and can be a challenging and debilitating diagnosis [2, 3]. While relatively rare in the Western Hemisphere, the incidence of NPC increases with age and is most prevalent among Southeast Asian and Mediterranean populations [4]. In the United States, incidence of NPC is low and disproportionately affects African Americans under the age of 20 [5]. Moreover, risk factors for NPC include occupational hazards such as cotton dust and dye in addition to salted fish [6, 7]. Importantly, infection of the nasopharynx with the Epstein Barr Virus (EBV) confers the greatest risk for developing NPC and represents the majority of pediatric NPC cases [8, 9]. In general, treatment for NPC primarily consists of radiotherapy with concomitant chemotherapy, which are effective in prolonging survival. However, despite favorable outcomes, relapse rates are relatively high and late effects of these treatment modalities can cause consequential morbidity [10, 11].

Typically, NPC is poorly differentiated and presents at a more advanced stage in children when compared to adult onset NPC [10, 12]. Even in general, a challenging characteristic of NPC is that it does often present in an advanced stage due to its anatomical origin [13]. There are three types of NPC characterized by histological features [14, 15]; type 1 (squamous cell), type 2 (non-keratinizing), and type 3 (undifferentiated) [16]. While most evidence suggests EBV is associated with types 2 and 3 NPC tumors, recent studies suggest a prominent role in type 1 tumors. The pathophysiology of EBV leading to the oncogenesis of NPC has not been completely elucidated; however, some have suggested that its tumorigenesis is triggered by latent membrane proteins (LMP1, LMP2A, and LMP2B) and EBV-determined nuclear antigens (EBNA1 and EBNA2) [17].

Specifically, LMP has been shown to promote NPC tumor development in mouse models and is thought to be driving factor in human NPC [18]. LMP has been shown to inhibit apoptosis and induce epithelial-mesenchymal transition (EMT) in NPC cells, and together with p16 expression, was associated with poorer outcomes and overall prognosis [15, 19]. However, canonical pathways associated with NPC pathogenesis are not entirely established. In particular, the *wnt* pathway is important for NPC development as it facilitates nuclear accumulation of β-catenin and coordinates cellular EMT and tumor oncogenesis [15, 20, 21]. There are also a variety of other factors that play a role in the development of NPC. Of note, homeobox A10 (HOXA10), activating transcription factor 1 (ATF1), and Interleukin 1-B (IL-1B) polymorphisms have been reported to be associated with NPC, among a multitude of others [22–24].

Further studies are required to better understand molecular mechanisms involved in NPC pathophysiology, such as the roles of lipopolysaccharide (LPS) and general inflammation on

nasopharyngeal tissue injury, HOXA10 and matrix metalloproteinases 1 and 3 (MMP1, MMP3) pathways. LPS is a bacterial endotoxin on gram negative bacteria and makes contact with the nasopharynx through normal airflow and eating that can lead to the induction of numerous pathological pathways associated with immune defense [25, 26]. As such, it has been hypothesized that LPS could be a leading driver and inducer of tumorigenesis of NPC, possibly through inflammation [26]. However, the relationship has not been completely elucidated to date and needs more investigation. As such, bioinformatics and performing large scale analysis on publicly available tissue samples is a reasonable avenue to pursue to shed light on this potential relationship.

Meta-analysis of transcriptome data permits more accurate investigation of disease-wide associations between different genes or gene sets. Unearthing common genetic contributions may provide new targets for existing therapeutic interventions while highlighting potential prognostic biomarkers that could be integrated in the management of NPC to improve outcomes and reduce recurrence rates.

The National Center for Biotechnology Information's (NCBI) Gene Expression Omnibus (GEO) is an open-access library consisting of thousands of biological samples from experimentation [27–29]. The Search Tag Analyze Resource for GEO (STARGEO) was developed by authors of this investigation to aggregate and perform meta-analyses on GEO transcriptomic data and has been validated based on similarity to The Cancer Genome Atlas (TCGA) Pan-Cancer dataset [28]. This study aims to use this technology to elaborate on purported mediators of NPC pathophysiology and determine potential alternative avenues for therapeutic targeting.

## Materials and methods

We employed the STARGEO platform to conduct a meta-analysis using the NCBI GEO repository to define NPC pathogenesis. More information on Gene Expression Omnibus can be found in the original reporting [27]. STARGEO is a robust tool to easily tag samples from studies on GEO and perform the appropriate statistical measures for users. Additionally, the large dataset of NPC samples made STARGEO, along with its user-friendliness, the ideal tool to investigate NPC. More information can be found in our previous paper [28]. Briefly, we used a random effect model for meta-analysis was used to produce effect sizes and meta p-values. We used a variation of the inverse-variance method to incorporate an assumption that the different studies are estimating different, yet related, intervention effects. To undertake a random-effects meta-analysis, the standard errors of the study-specific estimates are adjusted to incorporate a measure of the extent of variation, or heterogeneity, among the intervention effects observed in different studies (this variation is often referred to as tau-squared ($\tau2$, or Tau2)). The amount of variation, and hence the adjustment, can be estimated from the intervention effects and standard errors of the studies included in the meta-analysis [30]. Additionally, we scaled the fold change of each gene's effect by the significance ($-\log10$(P-value) × fold change) and used this score to rank genes by their differential expression and estimate the overlap among the top 200 (1%) of genes shared between the two datasets (NPC/control).

We tagged 111 NPC samples and 43 samples of healthy nasopharyngeal epithelium as controls across five independent studies. We used samples from the GSE12452 [31], GSE13597 [32], GSE53819 [33], GSE40290 [34], and GSE64634 [35] series (Fig 1). More information on the samples tagged within the studies can be found at http://stargeo.org/analysis/612/. NPC samples were taken at time of diagnosis and prior to any treatment. Tagged samples were from EBV-associated NPC tumor samples.

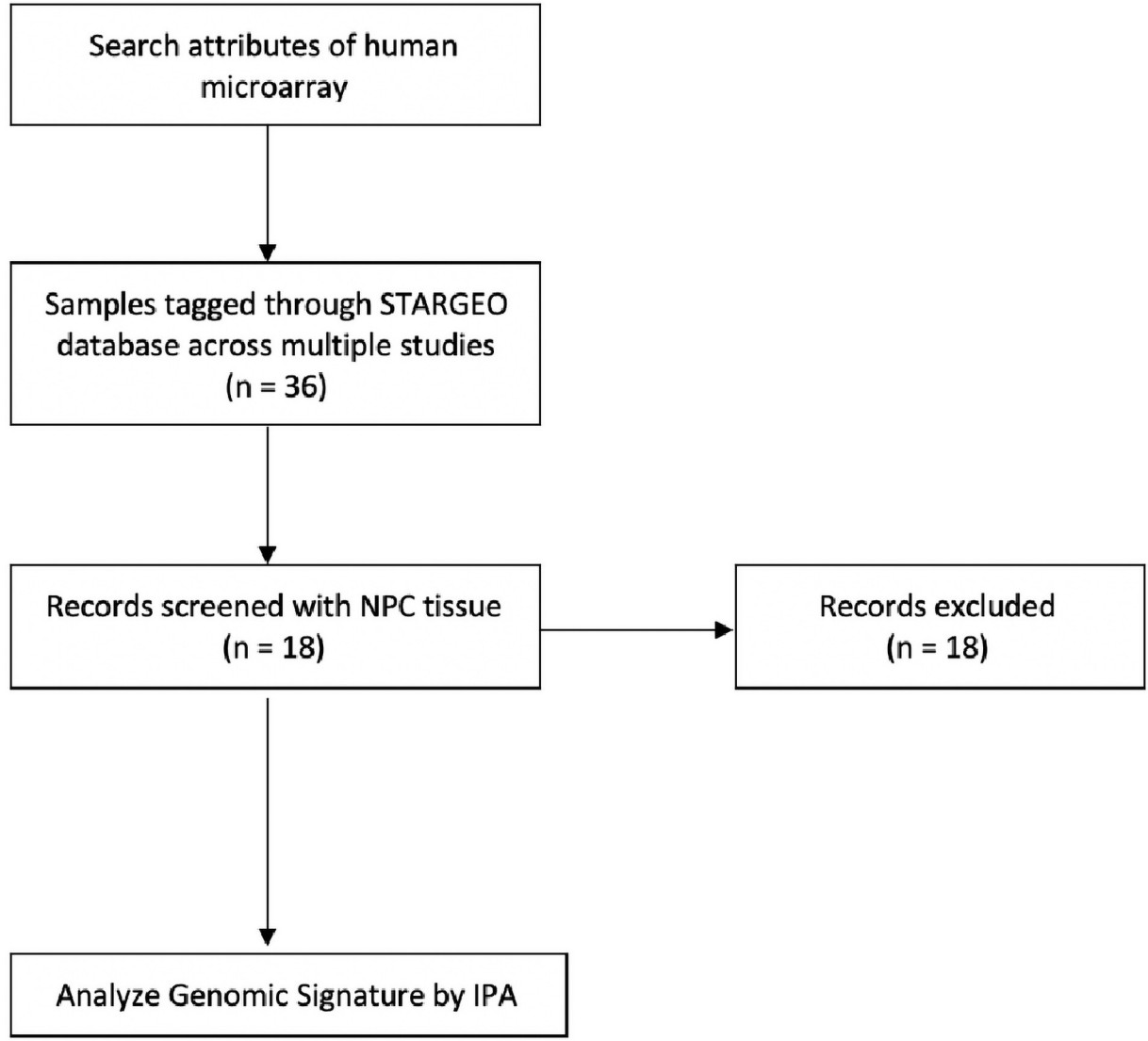

**Fig 1. Modified PRISMA diagram of STARGEO sample selection.**

We extracted 21,534 genes from our meta-analysis that was then analyzed through the Ingenuity Pathway Analysis (IPA) software. We restricted analyzed genes that showed statistical significance (p<0.05) and an absolute experimental log ratio, base of 2, greater than 0.15 between disease and control samples. IPA allows users to bring biological context to omics data, identify canonical pathways in a dataset, and build experimental models to better understand disease. The utility of IPA is expanded on here [36]. A total of 1,510 genes were included in the IPA analysis. This ultimately allowed us to expound upon the process of NPC and potential signatures and theorize on potential drug targets (Table 1). IPA identifies biological "entities" including canonical pathways, curated pathways of well-established metabolic pathways, and top upstream regulators, genes that IPA identifies as having the most influence on effecting the up or downregulation of genes in the inputted dataset. IPA calculates the p-values for these entities using the Fisher's Exact Test that compares the inputted dataset to the ~52,000 "entities" in IPA's database. IPA also produces a z-score, which predicts both

**Table 1. Top upregulated and downregulated genes between normal and NPC tissue along with their corresponding experimental log ratios.**

| Upregulated Genes in NPC (log ratio) | | Downregulated Genes in NPC (log ratio) | |
|---|---|---|---|
| HOX A10 | 0.814 | SNTN | -0.619 |
| CSAG 2/3 | 0.610 | LTF | -0.613 |
| FERMT1 | 0.474 | c7orf57 | -0.609 |
| PKP1 | 0.449 | MUC16 | -0.527 |
| IGF2 B3 | 0.427 | c9orf135 | -0.483 |
| MMP3 | 0.395 | EFCAB1 | -0.483 |
| MMP1 | 0.375 | TMEM232 | -0.475 |
| PTG S2 | 0.372 | c11orf97 | -0.457 |
| MMP12 | 0.192 | ADH 1B | -0.453 |
| VEGFA | 0.127 | CFAP 52 | -0.449 |

significance and activation state [36]. Z-score is ultimately calculated using the equation $z = \frac{N_+ - N_-}{\sqrt{N}}$ where N represent the entity, N+ correctly matching entities between the inputted dataset and the "entities" in IPA's database and N- representing the incorrect matches. The *White Paper* goes into greater detail on how IPA calculates p- and z-values for canonical pathways and upstream regulators [37]. We used IPA upstream analysis to screen for drugs that could have potential use in NPC. No IRB approval was deemed necessary for this study as there was no direct involvement of human subjects or patient-protected information.

## Results

### Top up and down regulated genes

We conducted meta-analysis using STARGEO on a total of 111 NPC tumor samples, using 43 control samples as control, to identify key gene and disease pathways in NPC pathogenesis. Results were processed in Ingenuity Pathway Analysis to provide biological context to our raw data. From IPA analysis, we identified the hepatic stellate cell activation and fibrosis as the top canonical pathway (p = 1.62E-08). Additionally, granulocyte adhesion and diapedesis (p = 4.11E-08), agranulocyte adhesion and diapedesis (p = 1.30E-07), glycoprotein VI (GP6) signaling pathway (p = 1.53E-07), and atherosclerosis signaling (p = 3.54E-07) were other top canonical pathways.

We next focused on the top upregulated genes from our analysis. We identified overexpression of fermitin family homolog 1 (FERMT1, experimental log ratio = 0.474), plakophilin-1 (PKP1, experimental log ratio = 0.449), and matrix metalloproteinase-3 (MMP-3, Fig 2). The highest up-regulated gene related to NPC tumorigenesis was transcription factor HOXA10 (experimental log ratio = 0.814), which is known to promote NPC development through interaction with the ZIC2 promoter [22].

Genes associated with extracellular matrix degradation were identified as important factors of NPC tumorigenesis. Specifically, the metalloproteinases MMP-1 and MMP-3 were upregulated (experimental log ratio = 0.395 and 0.375, respectively). MMP-1 has been extensively studied and is reported to be an essential factor in NPC development, primarily due to its interaction with the LMP-1 gene [16]. Likewise, MMP-3, an activator of MMP-1, has strong associations with NPC development and LPS-induced tissue injury, according to our analysis (Fig 2) [38]. Other oncogenic factors implicated in NPC development were also found to be upregulated including insulin-like growth factor 2 mRNA binding protein 3 (IGF2BP3, experimental log ratio = 0.427) and PTSGS2 (COX-2, experimental log ratio = 0.372).

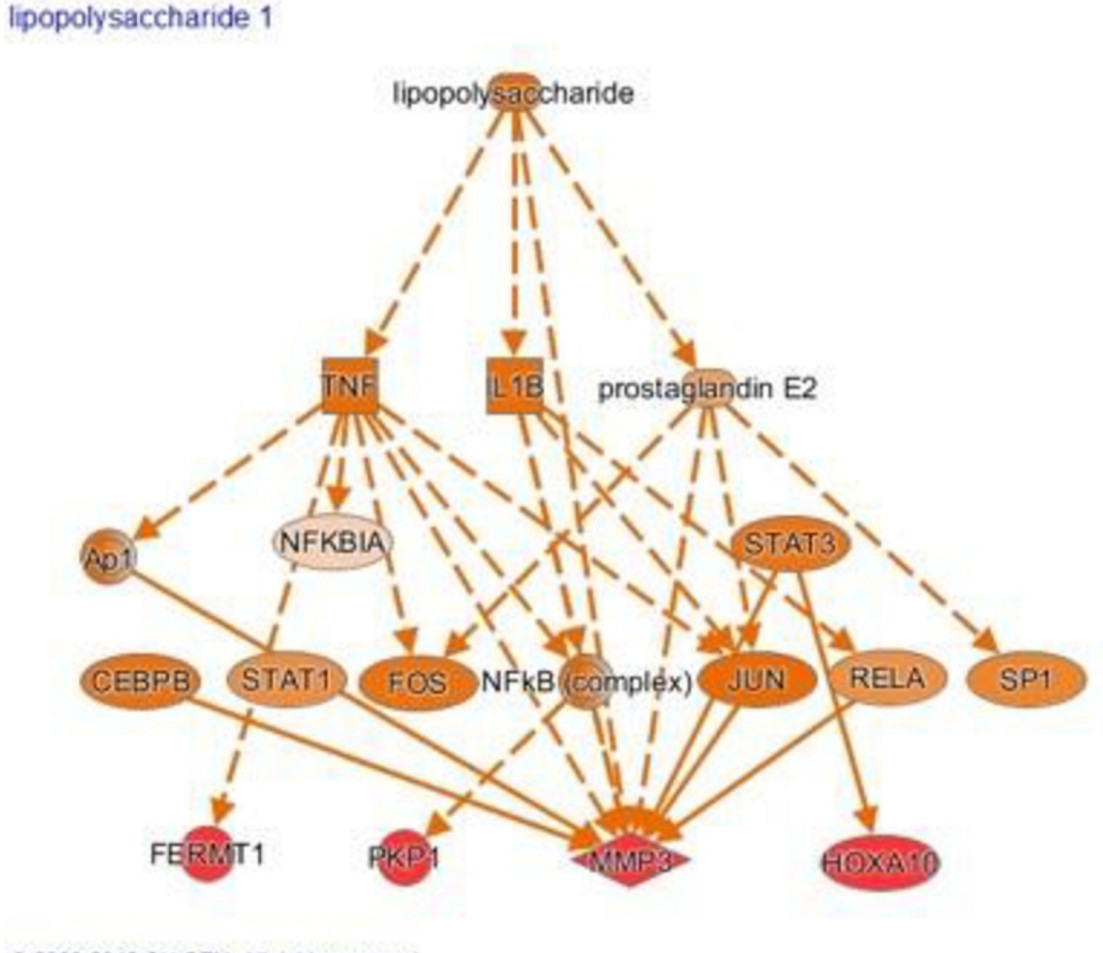

**Fig 2. LPS associated with the activity of many genes upregulated in NPC.** Image of the downstream analysis of etanercept by STARGEO platform showing inhibition of many LPS-mediated gene dysregulation.

Among the most downregulated genes in NPC tissue were those associated with having differentiated nasal epithelial cells in the tissue sample, indicating possible functional and structural mucosal integrity loss. Of these, sentan-cilia apical structure protein (SNTN) and lactotransferrin (LTF) were the most downregulated (experimental log ratio = -0.619 and -0.613, respectively).

## Causal analysis and upstream regulators

IPA Upstream Regulator Analysis recognizes molecules that best explain the observed gene expression in our dataset based on current knowledge of these molecules' effects on their target genes [36]. IPA examines the direction of change of target genes in our dataset compared to what is known and will either predict that the upstream regulator is activated (gene expression is mostly congruent with the regulator's known effects) or inhibited (gene expression is mostly incongruent with what is known). The z-score is a measure of the degree of activation, positive score, or inhibition, negative score. IPA identified lipopolysaccharide (LPS)-induced tissue injury as the top upstream regulator (p = 3.03E-14, activation Z-score = 4.507). LPS is a proxy

for inflammation and is associated with NPC carcinogenesis [32, 33]. For example, LPS stimulates multiple inflammatory pathways through toll-like receptor 4 (TLR4), leading to expression of prostaglandin E2 along with tumor necrosis factor (TNF) and IL-1B, as shown in our analysis (Fig 2). In keeping with these results, we found interluekin-1B (IL-1B, p = 1.18E-11, activation Z-score = 3.553), chromobox protein homolog 5 (CBX5, p = 1.38E-11, activation Z-score = 2.646), and interferon-gamma (IFN-G, p = 3.39E-11, activation Z-score = 2.890) were strongly activated whereas the small molecular inhibitor SB203580 (p = 1.64E-12, activation Z-score = -4.673) was predicted to be inhibited. As expected, these upstream regulators are all associated with NPC pathogenesis [24, 39–42]. Importantly, through IPA we found a link between LPS and IL-1B activation (Fig 2).

## Therapeutic target analysis

IPA Upstream Regulator can also be used to propose potential new drug candidates. IPA identifies drugs as upstream regulators, as described above. A negative z-score would entail that the genes in the dataset are incongruent with the direction that the drug would have on those genes. Thus, a drug with a negative z-score can potentially "correct" the genetic aberrations identified in our dataset. In search of new indications of existing drugs for the treatment of NPC we screened drug candidates with the highest negative z-scores. After we identified candidate drugs, we looked at the downstream genes known by IPA and looked at the total number of genes in that set that were upregulated in our analysis but would be downregulated by the drug. Infliximab and etanercept had the highest negative activation Z-score, indicating that it could downregulate its downstream genes that were upregulated in our analysis (Tables 2 and 3). Therefore, these drugs may serve as potential candidates for treating NPC, especially as it relates to inflammation-induced injury. Of note, etanercept demonstrated marked inhibition

**Table 2. IPA analysis table of upregulated genes both in LPS-induced damage, NPC and inhibited by etanercept.**

| Upregulated in NPC | Upregulated in LPS | Inhibited by Etanercept |
|---|---|---|
| HOX A10 | ✓ | * |
| CSAG 2/3 | * | * |
| FERMT1 | ✓ | * |
| PKP1 | ✓ | * |
| IGF2 B3 | * | * |
| MMP3 | ✓ | *Possible |
| COL 11A1 | * | * |
| KREMEN2 | * | * |
| MMP1 | ✓ | ✓ |
| PTG S2 | ✓ | ✓ |

**Table 3. IPA analysis table of gene names that are upregulated in LPS-induced damage, NPC tissue, and inhibited by infliximab.**

| Upregulated in NPC | Upregulated in LPS | Inhibited by Infliximab |
|---|---|---|
| IL1B | ✓ | ✓ |
| PTGS2 (COX-2) | ✓ | ✓ |
| VEGFA | ✓ | ✓ |
| MMP12 | ✓ | ✓ |

of genes involved in LPS-mediated tissue injury and possibly a role in the process of transformation of normal epithelial cells to cancer cells, such as MMP-1 and PTGS2 (Fig 3). Additionally, infliximab was predicted to inhibit multiple Akt pathway signaling components as well as downstream mediators including PTGS2 and IL-1B (Fig 3).

## Discussion

### Canonical pathways

Through meta-analysis of publicly available patient samples via STARGEO and IPA, we identified hepatic stellate cell activation and fibrosis as the top canonical pathway. Induction of Wnt pathway components result in nuclear accumulation of B-catenin, an important mediator of EMT and NPC tumorigenesis [43].

### Common genes

In support of prior literature, we found similar overexpression in NPC tissue compared to normal nasal epithelium in multiple different genes (Fig 4) such as HOXA10, which was the most upregulated gene in NPC tissue. HOXA10 has been shown to promote development of NPC through its interaction with ZIC2, a protein closely tied to NPC development, potentially through inhibiting apoptosis of cancerous cell lines [44]. In addition, this analysis shows an overexpression of MMP-1, which has a strong role in the development of NPC, potentially by interacting with LMP1 [45, 46]. We also found overexpression of PTSG2, VEGFA, IL1B (Fig 4). Similarly, our analysis found decreased expression of genes that have been supported by NPC studies such as LTF, which has been shown to have an anti-tumor and anti-metastasis effects in NPC through suppressing Akt [47]. In addition, we found under-expression of C7orf57, a gene that is expressed in ciliated nasal epithelial cells and has been downregulated with head and neck cancer and specifically NPC [48].

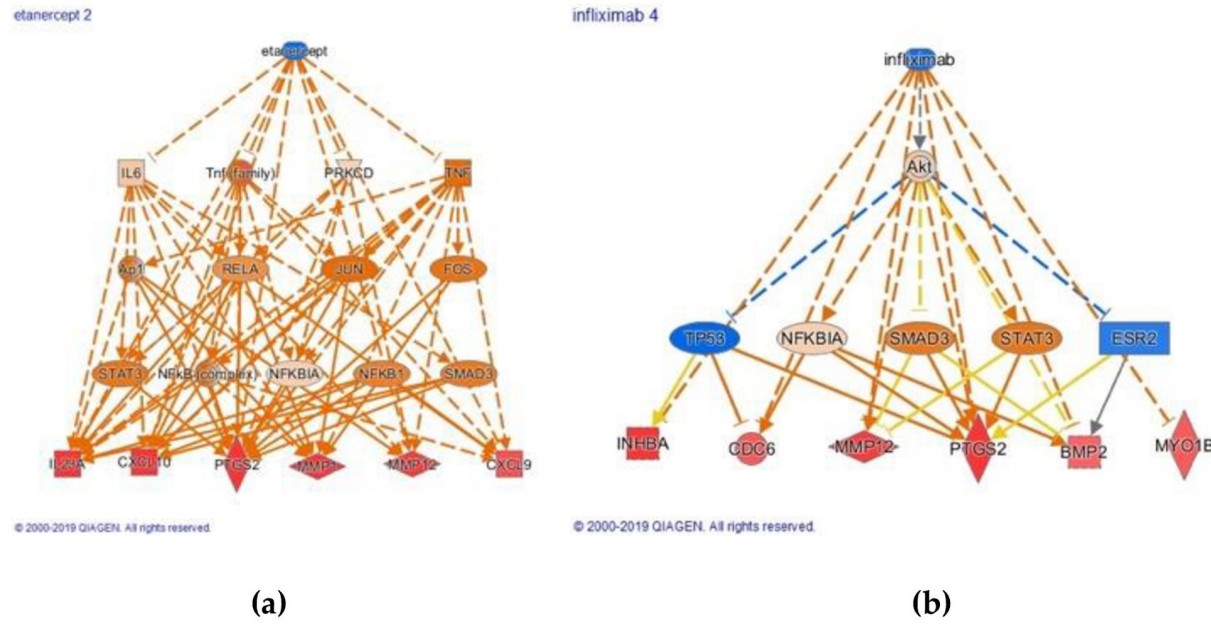

**(a)** **(b)**

**Fig 3.** a-b. IPA analysis and mapping of Etanercept (a) and Infliximab (b).

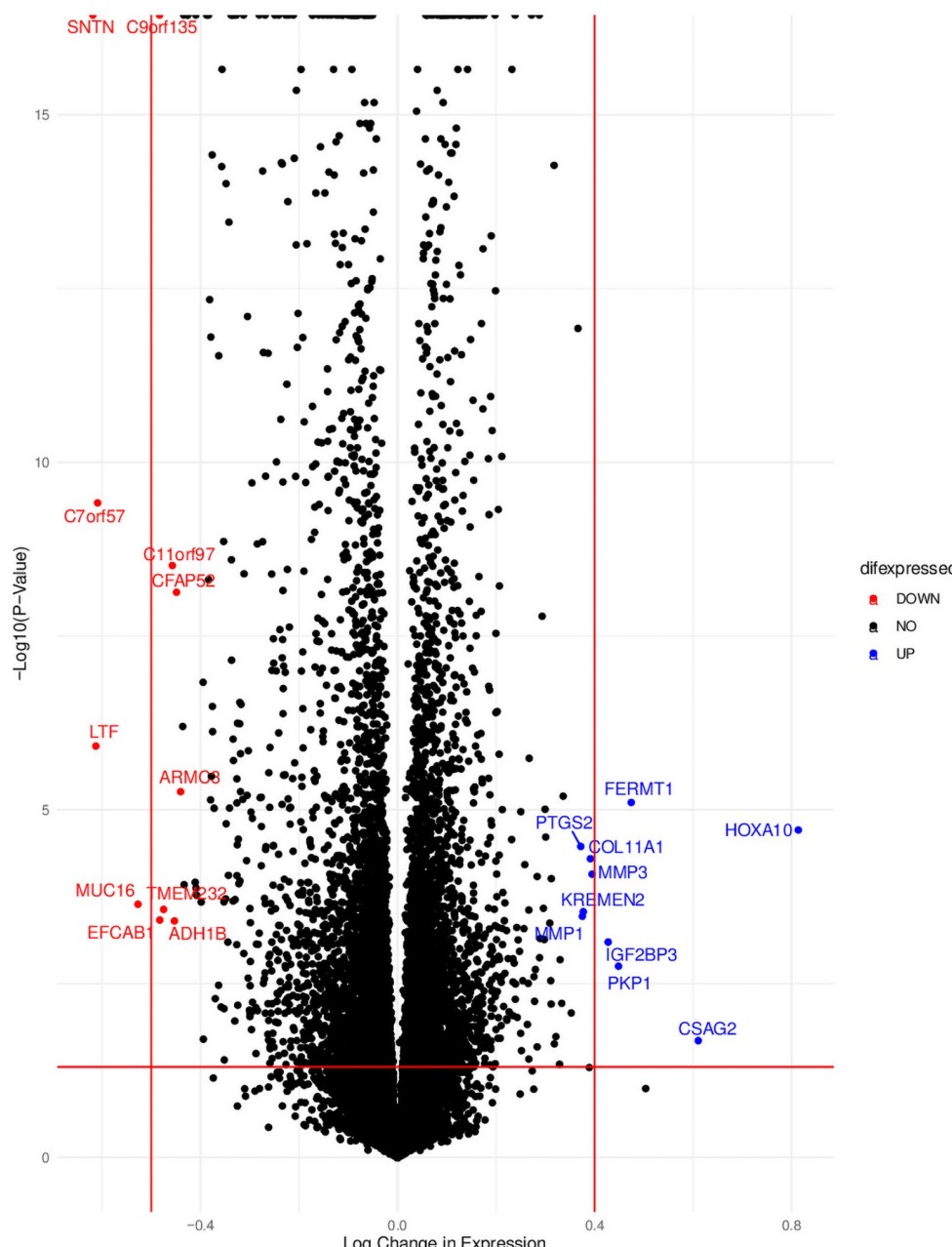

**Fig 4. A volcano plot illustrating the up and downregulated genes in NPC tissue versus normal tissue.** The y-axis is the -Log10(P-Value) illustrating significance while the x-axis is the Log change in expression, illustrating the effect difference between NPC tissue and normal tissue.

## Upstream regulators

As stated earlier, this meta-analysis predicted activation of LPS as a proxy for inflammation in NPC tissue. Inflammation has long been known to be associated with tumorigenesis–either as a response to tumor growth or a driver [49]. Specifically, multiple theories exist regarding LPS-induced inflammation and NPC oncogenesis, one of which involving repression of a protective gene; the short palate, lung and nasal epithelium clone 1 (SPLUNC1) [50]. LPS is also a

known inducer of the mitogen activated protein kinase (MAPK) pathway which enhances cellular proliferation in response to interleukin-1 (IL-1), another oncogenic factor for NPC [51]. Additionally, the nasopharynx anatomic location results in constant exposure to metabolites like LPS from the host microbiome. Thus, LPS-containing gram negative bacteria are thought to stimulate TLR4-dependent inflammation by resident macrophages, resulting in the secretion of cytokines (such as TNF and IL-1), which irritates the mucosa and induces MAPK-mediated cellular proliferation and tumorigenesis [52]. Indeed, LPS is a key aspect in our analysis, as it activates TNF, IL-1B, and prostaglandin E2, all three of which are known contributors to NPC pathogenesis [40, 53]. Additionally, our analysis relates LPS activity to induction of FERMT1, PKP1, HOXA10, and MMP-3, a protein inseparably responsible for extracellular matrix breakdown and NPC development. Thus, inhibitors that mitigate LPS-induced damage of the nasopharyngeal epithelium or target downstream components such as TNF and MMP-3 could be beneficial for preventing possible inflammation-triggered tumorigenesis.

In addition to its pro-inflammatory properties, LPS is known to directly interact with IL-1B, a cytokine important for NPC development, which was profoundly upregulated in our analysis. For Chinese populations, there exists a reported association between IL-1B polymorphisms and NPC development [24]. In addition, NPC-infiltrating T cells acquire IL-1 secretory function, further driving tumorigenesis [54]. Furthermore, IL-1 has been shown to be produced by EBV-infected epithelial cells, inducing cellular growth via MAPK in a paracrine and autocrine fashion [55]. In particular, MAPK demonstrated strong associations in our analysis. In fact, we found that many MAPK-associated signaling components were upregulated in NPC samples while SB203580, a small molecule MAPK inhibitor, was predicted to be inhibited; activity of these factors are known mediators of NPC tumorigenesis [56]. Specifically, SB203580 has been shown to down-regulate the p38 MAPK pathway, resulting in MMP-2 inhibition, another protein important for extracellular matrix breakdown, ultimately limiting NPC invasion [57, 58].

Similar to IL-1, interferon-gamma (IFN-G) also stimulates the MAPK pathway and was upregulated in our analysis [59]. It is of note that depending on the cancer, IFN-G activity can either be oncogenic or tumor suppressive [60]. In the context of NPC, IFN-G polymorphisms (13-CA repeats) is linked with enhanced metastatic potential in some Northern Chinese populations [42]. In addition, targeting IFN-G has been posed as a potential therapeutic strategy for NPC, but warrants further investigation [61]. Our analysis suggests that IFN-G upregulation could be related to NPC tumorigenesis, possibly due to its promotion of MAPK signaling.

## Top up-regulated genes

As mentioned previously, MMP-1 and MMP-3 are involved in the physiologic breakdown of extracellular matrix which is essential for embryonic development, reproduction, and tissue remodeling. However, aberrant expression of these metalloproteinases is associated with various disease processes, including metastasis through enabling ease of spread and invasion. We show these aspects are highly elevated in our analysis. Specifically, our results support previous findings and suggests that in addition to MMP-1, MMP-3 could be involved in NPC tumorigenesis, thus supporting the clinical utility of anti-MMP therapeutic agents for NPC.

Our analysis also found FERMT1 highly upregulated in NPC samples. This protein is involved with integrin signaling and actin-extracellular matrix interactions. As with the other genes mentioned, FERMT1 has reported associations with cancer, likely through B-catenin regulation [62]. Similarly, we found upregulation of PKP1, an integrin involved with cell desmosomes linking cadherins to cytoskeletal intermediate filaments. Interestingly, PKP1 is reported to be a tumor suppressor but not yet been described in NPC [63].

## Top downregulated genes

The most downregulated genes, SNTN, MUC16 and ADH1B, are those associated with nasal epithelium maintenance, suggesting that a decrease in the epithelial barrier integrity could be involved in NPC development. Specifically, SNTN, a gene that encodes the protein sentan, was decidedly downregulated in our analysis. Sentan has been shown to be a part of the ciliary tip, connecting cell membrane with microtubules, and thus potentially associated with airway clearance [64, 65]. Normally, sentan is pointedly expressed in healthy nasopharyngeal tissue [64]. To date, its relationship with oncogenesis has not been well established however there have been fewer differentiated cells in a tumor than in a normal cell in prior investigations [65].

MUC16 is a mucin for which CA-125 is a repeating peptide. CA-125 is a well-known tumor marker in ovarian cancer and it is known to support proliferation and tumorigenesis [66]. However, MUC16 is also expressed in respiratory epithelial cells [67]. MUC16 has been a reported target for cancer therapy [68]. It has a reported association with NPC likely through EBV infection and has been theorized to be a tumor marker for NPC [69].

Last, ADH1B is a member of the alcohol dehydrogenase family and it is a gene that codes for proteins that metabolize a wide variety of substrates, including ethanol, retinol, other aliphatic alcohols, hydroxysteroids, and lipid peroxidation products. Polymorphisms of ADH1B have been associated with decreased risk of head and neck cancer in patients of Asian descent [70]. However, few studies have looked at deregulation of ADH1B expression or activity in the context of NPC. This analysis suggests that it is downregulated in NPC tissue compared to normal nasal epithelium.

## Therapeutic targets

Given what is known about the LPS pathway, it can be mapped out to further understand how it could possibly be related to oncogenesis through mediating inflammation. This would potentially provide alternatives for non-radiotherapy or chemotherapies such as etanercept and infliximab. Etanercept is a fusion protein that inhibits TNF and has been successfully used to treat a variety of inflammatory disorders, namely severe rheumatoid arthritis (RA) [71]. Of note, LPS-induced tissue injury is thought to be a major factor in the development of RA and possibly even osteoarthritis (OA), thus a common pathway for NPC and RA/OA exists [72, 73]. Despite its clinical efficacy in treating inflammatory-based diseases, little research into the possible effect of etanercept has been conducted in treating head and neck cancer such as NPC, which has an abject pro-inflammatory basis. Indeed, IPA analysis illustrated that common upregulated factors in NPC pathogenesis, especially those involved in LPS-mediated tissue damage such as MMP-1 and COX-2, are strongly inhibited by etanercept. Our analysis suggests that etanercept is known to lead to changes of expression in genes that were found to differentiate NPC tissue from normal epithelium.

Infliximab is another anti-TNF medication and poses an alternative possible therapeutic option through inhibition of the LPS pathway. Not only does it inhibit a multitude of pathways that were upregulated in NPC tissue, such as IL-1B and COX-2, it may also modulate IFN-G (as reported in the literature and in our analysis) and Akt-mediated signaling, both of which are related to poor prognosis in NPC [74]. Importantly, VEGFA has been shown to be upregulated in NPC invasion in our analysis (log ratio of 0.127) and in previous studies. Thus an inhibition of that, such as shown here by infliximab, may prove to be beneficial [75]. Another target of interest is MMP-12, which, while shown in preceding research and in our analysis (log ratio 0.192) to be upregulated in NPC tissue, was also potently inhibited by infliximab in our analysis.

To our knowledge, this is the first time etanercept and infliximab have been suggested as a possible treatment for NPC. However, their effectiveness requires clinical correlation and prospective evidence.

### Risk of oncogenic potential of etanercept and infliximab

There has been much discussion and controversy about the possibility of anti-TNF medications causing cancer [76, 77]. Specifically, TNF has been known to have multiple associations with carcinogenesis as both a possible enhancer of tumor growth and tumorigenesis in addition to possibly exerting an anti-tumorigenic effect [78].

A meta-analysis from Askling et al. in 2011 [79] suggests that other than non-melanoma skin cancer, both etanercept and infliximab (in addition to Adalimumab) did not lead to an increased risk of cancer. However, with non-melanoma skin cancer, the risk was increased. Another meta-analysis, this time from Leombruno et al. 2008, found no increased risk of lymphoma or non-melanoma skin cancer [77]. In both papers, the authors discussed in great length the difficulty of analyzing the available data (due to variations of reporting by the authors in the studies analyzed and differential grouping of cancers) and to take prudence when interpretating conclusions. In 2012, a review of the epidemiological studies into anti-TNF therapies was undertaken to shed light on the available evidence at that time [80]. The authors recommended more observational studies into the risks of anti-TNF therapy. Specifically, they found that only 11 studies at that time met their classification for observational studies and those that did differed significantly in methodology and allotment. Since then, a recent large cohort study from Sweden, completed by Raaschou et al. 2018 [81] also showed no increased risk of cancer in patients who were treated with adalimumab, certolizumab pegol, etanercept, golimumab, or infliximab. Another study, this time by Waljee et al. 2020 [82], investigated the risk of developing cancer with anti-TNF therapy in patients who had a history of cancer. This study found no increased risk of utilizing anti-TNF therapy [82]. Additionally the European Congress of Rheumatology published the findings of an investigation in 2019 that also found no increased risk of malignancy in patients with psoriatic arthritis treated with anti-TNF therapy in 4 nordic countries (Denmark, Finland, Iceland and Sweden) [83]. In conclusion, with recent data suggesting a relatively low risk of increased risk of developing cancer due to anti-TNF therapy, patients and care providers should feel assured when prescribing these medicines however stay vigilant of any future investigations into this topic.

### Limitations

Given that the data incorporated into this analysis is from a public source, our investigation is limited by patient characteristics such as co-morbidities, age, gender, and others. Thus, there is an inherent obstacle when evaluating heterogeneity in the gene effects given the technology.

### Conclusions

Through our analysis of over 150 pooled normal and NPC samples, we found that NPC tissue was associated with an overexpression of genes associated with degradation of the nasal epithelial barrier and LPS-induced inflammation and tissue damage compared to healthy nasal epithelium. Overall, crosstalk involving LPS, cytokines, and MMPs converge on inflammation, resulting in predicted inhibition by etanercept and infliximab. Further characterization of downstream targets of etanercept and infliximab may clarify its therapeutic role in the management of NPC. Ultimately, this analysis supports a hypothesis investigating whether or not LPS induces NPC initiation and oncogenesis and future prospective analyses should evaluate this topic given the potential for therapeutic advances.

## Supporting information

**S1 Checklist. PRISMA 2009 checklist.**
(DOC)

## Acknowledgments

The authors would like to acknowledge multiple people for their assistance in this project. Specifically, we would like to acknowledge Dr. Edmund Mroz, Dr. Ricardo Carrau, Dr. Dukagjin Blakaj for their insights and helpful advice in preparation of this manuscript.

## Author Contributions

**Conceptualization:** David Z. Allen, Jihad Aljabban, Dustin Silverman, Sean McDermott.

**Data curation:** David Z. Allen, Jihad Aljabban, Dexter Hadley.

**Formal analysis:** David Z. Allen, Jihad Aljabban.

**Investigation:** David Z. Allen, Jihad Aljabban, Dustin Silverman, Sean McDermott, Ross A. Wanner, Michael Rohr, Maryam Panahiazar.

**Methodology:** David Z. Allen, Jihad Aljabban, Dexter Hadley, Maryam Panahiazar.

**Project administration:** Dustin Silverman, Maryam Panahiazar.

**Resources:** Dexter Hadley, Maryam Panahiazar.

**Software:** Jihad Aljabban, Dexter Hadley.

**Supervision:** Dustin Silverman, Dexter Hadley, Maryam Panahiazar.

**Visualization:** David Z. Allen.

**Writing – original draft:** David Z. Allen, Jihad Aljabban, Dustin Silverman, Sean McDermott, Ross A. Wanner.

**Writing – review & editing:** David Z. Allen, Jihad Aljabban, Dustin Silverman, Sean McDermott, Ross A. Wanner, Michael Rohr, Maryam Panahiazar.

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
