## [Decision Letter · Decision Letter 0]

27 Apr 2021

PONE-D-20-37466

Meta-Analysis illustrates possible role of lipopolysaccharide

(LPS)-induced tissue injury in nasopharyngeal carcinoma (NPC) pathogenesis

PLOS ONE

Dear Dr. Allen,

Thank you for submitting your manuscript to PLOS ONE. After careful consideration, we feel that it has merit but does not fully meet PLOS ONE’s publication criteria as it currently stands. Therefore, we invite you to submit a revised version of the manuscript that addresses the points raised during the review process.

Both reviewers raised some pertinent and important issues that are related to description of the methodology used and heterogeneity of samples among others which need to be addressed.  

We look forward to receiving your revised manuscript.

Kind regards,

Srinivas Mummidi, D.V.M., Ph.D.

Academic Editor

PLOS ONE

Journal Requirements:

PLOS requires an ORCID iD for the corresponding author in Editorial Manager on papers submitted after December 6th, 2016. Please ensure that you have an ORCID iD and that it is validated in Editorial Manager. To do this, go to ‘Update my Information’ (in the upper left-hand corner of the main menu), and click on the Fetch/Validate link next to the ORCID field. This will take you to the ORCID site and allow you to create a new iD or authenticate a pre-existing iD in Editorial Manager. Please see the following video for instructions on linking an ORCID iD to your Editorial Manager account: https://www.youtube.com/watch?v=_xcclfuvtxQ

3. We note that this manuscript is a systematic review or meta-analysis; our author guidelines therefore require that you use PRISMA guidance to help improve reporting quality of this type of study. Please upload copies of the completed PRISMA checklist as Supporting Information with a file name “PRISMA checklist”.

4. We note that Figures 2-3 in your submission contain copyrighted images. All PLOS content is published under the Creative Commons Attribution License (CC BY 4.0), which means that the manuscript, images, and Supporting Information files will be freely available online, and any third party is permitted to access, download, copy, distribute, and use these materials in any way, even commercially, with proper attribution. For more information, see our copyright guidelines: http://journals.plos.org/plosone/s/licenses-and-copyright.

1.              You may seek permission from the original copyright holder of Figures 2-3 to publish the content specifically under the CC BY 4.0 license.

Reviewers' comments:

Reviewer's Responses to Questions

**Comments to the Author**

1. Is the manuscript technically sound, and do the data support the conclusions?

Reviewer #1: Yes

Reviewer #2: Partly

2. Has the statistical analysis been performed appropriately and rigorously? 

Reviewer #1: I Don't Know

Reviewer #2: Yes

3. Have the authors made all data underlying the findings in their manuscript fully available?

Reviewer #1: Yes

Reviewer #2: Yes

4. Is the manuscript presented in an intelligible fashion and written in standard English?

Reviewer #1: Yes

Reviewer #2: Yes

5. Review Comments to the Author

Reviewer #1: Allen et al. carried out a meta-analysis pointing to a possible role of LPS-induced tissue injury in NPC pathogenesis. The authors also performed sophisticated IPA analysis. The statistical analysis was very poorly described. My criticisms are intended to increase the quality of the paper by presenting more detailed information regarding the statistical methods.

1) On page 3 at line 95 the authors stated “. . . standard and random fixed effects models for meta-analysis was used to produce effect sizes and meta p-values.” Prior to this statement the authors indicated that the STARGEO platform was used to conduct the meta-analysis and they also cited a previous paper from their group by Hadley et al. (2017) for more information (their reference 27). There are several problems in this regard that should be addressed:

A) It is unclear what “standard and random fixed effects models” means. Do the authors mean both effects together as in a mixed effects model or one or the other as dictated by the data across studies?

B) Citing the STARGEO platform does not suffice as a description of the meta-analysis approach. Inspection of the STARGEO website did not reveal any mention of exactly how the meta-analysis is carried out. Moreover, citing Hadley et al. (2017) for more information also does not suffice because that paper does not adequately describe the meta-analysis approach.

C) How is heterogeneity (if a random or mixed effects model was used) modeled?

D) How are effect sizes combined? The authors mentioned that the Dersimonian-Laird inverse variance method was used to study weight percentages. Was this used to combine effect sizes? How is error in the combined effect size estimated?

2) The authors quoted a number of p-values as being obtained from IPA analysis (e.g., page 4 at lines 125 to 127). Please describe the specifics of the test used by the IPA software to compute these p-values.

3) On page 5 at line 159, the authors mention the use of z-scores to measure the degree of activation (if positive) or inhibition (if negative). Is this a meta-analysis z-score? If so, the answer to 1D is relevant because it involves the standard error of the meta-analysis combined effect. If not, then please describe how this z-score is obtained.

Reviewer #2: This meta-analysis examined lipopolysaccharides (LPS) as an effector of nasopharyngeal carcinoma (NPC). The authors contrasted 111 tumor samples with 43 samples of normal epithelium. The study is original, including a very common (almost ubiquitous) risk: the exposure to LPS as a possible etiology for the disease. Nevertheless, the researchers have weak points that must be addressed to strengthen the manuscript:

1. In the introduction section, the authors briefly mentioned LPS. The authors must clarify mechanistic LPS paths to reach the nasopharynx and plausible physiopathology. Should NPC be related to local microbiota?

2. The analysis did not give any clue regarding the heterogeneity between samples in the STARGEO dataset. Can the authors compute the variance attributable to the tissue samples? Do they have access to clinical features of the tissue samples? The differentiation of the histological type of NPC is relevant due to its relation with the Epstein Barr virus. The NPC type can be a source of heterogeneity and limitation for the analysis.

3. It is essential to have a correct interpretation of the contrast between pathological vs healthy tissue. The authors should clarify which type of log-ratio function was computed. As this function is a binary form, some researchers divide the log2(NPC/control) by two, others no. If the authors used the straight log2 ratio, what was the rationale to use a cutoff point of 0.15? The highest value was less than 1, therefore the researchers worked with tiny effect sizes, meanwhile, large values on z-score came from the Ingenuity analysis.

4. The researchers used a standard (perhaps "fixed") and randomized meta-analysis. How did they adjust by confounders? Or they did not consider these adjustments? (Perhaps due to restriction from the original dataset). Please comment on the discussion.

5. The use of Etanercept or Infliximab can induce some cancers and both have constraints as therapeutic agents for tumors. The authors should comments in deep on the pros and cons.

6. A call for prudence regarding the reference on alcohol as a risk factor for NPC. The cited study gave weak evidence to this statement: It was a retrospective design, smoking was a big confounder (except for heavy drinkers) associated with alcohol consumption, and finally, the calculated HR showed a small effect. Looks that there are few commonalities between drinking molecular pathways (large effects) and the NPC (small support from the meta-analysis).

7. Finally, the described findings are common to many other pathologies and the proposed etiology, and the suggestion that there was a common risk factor between NPC and hepatic cirrhosis is alcohol does not have strong support.

6. PLOS authors have the option to publish the peer review history of their article (what does this mean?). If published, this will include your full peer review and any attached files.

Reviewer #1: No

Reviewer #2: No

---

## [Author Response · Author response to Decision Letter 0]

22 Aug 2021

Please see attached document, to be honest I am unsure what this question is asking/how it differs from the official response to the reviewers document that was uploaded.

To Whom It May Concern:

Thank you for your comments. We did our best to respond to all of them. We made the new manuscript reflect the changes by using red font. Please let me know if we need to alter anything. 

Thank you, we made the appropriate changes.

PLOS requires an ORCID iD for the corresponding author in Editorial Manager on papers submitted after December 6th, 2016. Please ensure that you have an ORCID iD and that it is validated in Editorial Manager. To do this, go to ‘Update my Information’ (in the upper left-hand corner of the main menu), and click on the Fetch/Validate link next to the ORCID field. This will take you to the ORCID site and allow you to create a new iD or authenticate a pre-existing iD in Editorial Manager. Please see the following video for instructions on linking an ORCID iD to your Editorial Manager account: https://www.youtube.com/watch?v=_xcclfuvtxQ

We linked an ORCID iD account. 

We note that this manuscript is a systematic review or meta-analysis; our author guidelines therefore require that you use PRISMA guidance to help improve reporting quality of this type of study. Please upload copies of the completed PRISMA checklist as Supporting Information with a file name “PRISMA checklist”.

We created a PRISMA guidance and attached it into the Supporting Information. 

4. We note that Figures 2-3 in your submission contain copyrighted images. All PLOS content is published under the Creative Commons Attribution License (CC BY 4.0), which means that the manuscript, images, and Supporting Information files will be freely available online, and any third party is permitted to access, download, copy, distribute, and use these materials in any way, even commercially, with proper attribution. For more information, see our copyright guidelines: http://journals.plos.org/plosone/s/licenses-and-copyright.

1. You may seek permission from the original copyright holder of Figures 2-3 to publish the content specifically under the CC BY 4.0 license.

Thank you for this comment, we have an IPA license and as such, we have a right to use their images. Please email me at davidzallen614@gmail.com if I can provide anymore assistance in this matter.

Reviewer #1: Allen et al. carried out a meta-analysis pointing to a possible role of LPS-induced tissue injury in NPC pathogenesis. The authors also performed sophisticated IPA analysis. The statistical analysis was very poorly described. My criticisms are intended to increase the quality of the paper by presenting more detailed information regarding the statistical methods.

1) On page 3 at line 95 the authors stated “. . . standard and random fixed effects models for meta-analysis was used to produce effect sizes and meta p-values.” Prior to this statement the authors indicated that the STARGEO platform was used to conduct the meta-analysis and they also cited a previous paper from their group by Hadley et al. (2017) for more information (their reference 27). There are several problems in this regard that should be addressed:

A) It is unclear what “standard and random fixed effects models” means. Do the authors mean both effects together as in a mixed effects model or one or the other as dictated by the data across studies?

Thank you for asking this question, though STARGEO generates produces both random and fixed effects, we only used the random model for our analysis and made the clarification.

B) Citing the STARGEO platform does not suffice as a description of the meta-analysis approach. Inspection of the STARGEO website did not reveal any mention of exactly how the meta-analysis is carried out. Moreover, citing Hadley et al. (2017) for more information also does not suffice because that paper does not adequately describe the meta-analysis approach.

C) How is heterogeneity (if a random or mixed effects model was used) modeled?

We designed a simple analytical where more advanced curators could design, compute and visualize standard genomic meta-analysis of both random and fixed effects across tagged and annotated digital samples. We also used spearman rank correlation across all comparisons of differentially expressed genes between STARGEO.org (random versus fixed effects) meta-analyses.

D) How are effect sizes combined? The authors mentioned that the Dersimonian-Laird inverse variance method was used to study weight percentages. Was this used to combine effect sizes? How is error in the combined effect size estimated?

This used to calculate the weight for estimation of the “random effects”. Specifically, we used inverse variance weighting for pooling of the data across studies with different sizes , and calculated weights for estimates of random effects with continuous outcome data via the DerSimonian-Laird estimate. A variation on the inverse-variance method is to incorporate an assumption that the different studies are estimating different, yet related, intervention effects. To undertake a random-effects meta-analysis, the standard errors of the study-specific estimates are adjusted to incorporate a measure of the extent of variation, or heterogeneity, among the intervention effects observed in different studies (this variation is often referred to as tau-squared (τ2, or Tau2)). The amount of variation, and hence the adjustment, can be estimated from the intervention effects and standard errors of the studies included in the meta-analysis.

2) The authors quoted a number of p-values as being obtained from IPA analysis (e.g., page 4 at lines 125 to 127). Please describe the specifics of the test used by the IPA software to compute these p-values.

Thank you for pointing this out as it was not clear. The canonical pathways and top upstream regulators are calculated by IPA in the cited paper in line 112. We added some clarification and the appropriate citations. 

3) On page 5 at line 159, the authors mention the use of z-scores to measure the degree of activation (if positive) or inhibition (if negative). Is this a meta-analysis z-score? If so, the answer to 1D is relevant because it involves the standard error of the meta-analysis combined effect. If not, then please describe how this z-score is obtained.

Thank you for this comment, IPA calculates the z-score with this software and we added how it is done in the manuscript. 

Reviewer #2: This meta-analysis examined lipopolysaccharides (LPS) as an effector of nasopharyngeal carcinoma (NPC). The authors contrasted 111 tumor samples with 43 samples of normal epithelium. The study is original, including a very common (almost ubiquitous) risk: the exposure to LPS as a possible etiology for the disease. 

Nevertheless, the researchers have weak points that must be addressed to strengthen the manuscript:

1. In the introduction section, the authors briefly mentioned LPS. The authors must clarify mechanistic LPS paths to reach the nasopharynx and plausible physiopathology. Should NPC be related to local microbiota?

Thank you for this comment. We added more information regarding the background of LPS and its exposure into the nasopharynx. 

2. The analysis did not give any clue regarding the heterogeneity between samples in the STARGEO dataset. Can the authors compute the variance attributable to the tissue samples? Do they have access to clinical features of the tissue samples? The differentiation of the histological type of NPC is relevant due to its relationship with the Epstein Barr virus. The NPC type can be a source of heterogeneity and limitation for the analysis.

Thank you for this comment. After looking through the underlying research, these NPC samples were all EBV-related. 

3. It is essential to have a correct interpretation of the contrast between pathological vs healthy tissue. The authors should clarify which type of log-ratio function was computed. As this function is a binary form, some researchers divide the log2(NPC/control) by two, others no. If the authors used the straight log2 ratio, what was the rationale to use a cutoff point of 0.15? The highest value was less than 1, therefore the researchers worked with tiny effect sizes, meanwhile, large values on z-score came from the Ingenuity analysis.

We scaled the fold change of each gene’s effect by the significance (−log10(P-value) × fold change), and used this score to rank genes by their differential expression and estimate the overlap among the top 200 (1%) of genes shared between the two datasets (NPC/control). 

4. The researchers used a standard (perhaps "fixed") and randomized meta-analysis. How did they adjust by confounders? Or they did not consider these adjustments? (Perhaps due to restriction from the original dataset). Please comment on the discussion.

Thank you for this comment. We clarified with reviewer comments that a random model was used. We were limited by the original dataset so could not adjust for confounding variables such as co-morbidities. 

5. The use of Etanercept or Infliximab can induce some cancers and both have constraints as therapeutic agents for tumors. The authors should comments in deep on the pros and cons.

Thank you for the comment. We discussed in detail the existing literature into the oncogenic potential of etanercept and infliximab. 

6. A call for prudence regarding the reference on alcohol as a risk factor for NPC. The cited study gave weak evidence to this statement: It was a retrospective design, smoking was a big confounder (except for heavy drinkers) associated with alcohol consumption, and finally, the calculated HR showed a small effect. Looks that there are few commonalities between drinking molecular pathways (large effects) and the NPC (small support from the meta-analysis).

Thank you for this comment. Given the underlying literature, we removed from paper. 

7. Finally, the described findings are common to many other pathologies and the proposed etiology, and the suggestion that there was a common risk factor between NPC and hepatic cirrhosis is alcohol does not have strong support.

Thank you for this comment. Given the underlying literature, we removed from paper.

---

## [Decision Letter · Decision Letter 1]

22 Sep 2021

Meta-Analysis illustrates possible role of lipopolysaccharide

(LPS)-induced tissue injury in nasopharyngeal carcinoma (NPC) pathogenesis

PONE-D-20-37466R1

Dear Dr. Allen,

We’re pleased to inform you that your manuscript has been judged scientifically suitable for publication and will be formally accepted for publication once it meets all outstanding technical requirements. Before your final submission, I would recommend addressing the suggestion of the second reviewer.

Kind regards,

Srinivas Mummidi, D.V.M., Ph.D.

Academic Editor

PLOS ONE

Additional Editor Comments (optional):

Reviewers' comments:

Reviewer's Responses to Questions

**Comments to the Author**

1. If the authors have adequately addressed your comments raised in a previous round of review and you feel that this manuscript is now acceptable for publication, you may indicate that here to bypass the “Comments to the Author” section, enter your conflict of interest statement in the “Confidential to Editor” section, and submit your "Accept" recommendation.

Reviewer #1: All comments have been addressed

Reviewer #2: All comments have been addressed

2. Is the manuscript technically sound, and do the data support the conclusions?

Reviewer #1: Yes

Reviewer #2: Yes

3. Has the statistical analysis been performed appropriately and rigorously? 

Reviewer #1: Yes

Reviewer #2: Yes

4. Have the authors made all data underlying the findings in their manuscript fully available?

Reviewer #1: Yes

Reviewer #2: Yes

5. Is the manuscript presented in an intelligible fashion and written in standard English?

Reviewer #1: Yes

Reviewer #2: Yes

6. Review Comments to the Author

Reviewer #1: The clarifications incorporated in the revised version have made the paper stronger and acceptable for publication. Congratulations on the outstanding work!

Reviewer #2: The authors accomplished the requested clarifications in concepts like LPS, EBV-related samples. The authors made a good analysis of the potential oncogenicity of the recommended drugs.

The authors dropped the concept of alcohol pathogenesis because it did not have support according to the current scientific literature.

The authors have problems dealing with the sources of heterogeneity of the sample. This is concerning the scarce information on data confounders in the specific platform they used. Like any other gene repository, it has its constraints. Strictly speaking, this paper is not like a literature Meta-Analysis. The PRISMA must be modified for making this specific genetic analysis.

The authors addressed the questions I had. The paper can be published with a minor clarification on the difficulty to analyze the source of heterogeneity in the gene effects, this difficulty is inherent to the platform limitation on including confounders, and then con not be implement the used statistical methodology.

7. PLOS authors have the option to publish the peer review history of their article (what does this mean?). If published, this will include your full peer review and any attached files.

Reviewer #1: No

Reviewer #2: No

---

## [Editor Report · Acceptance letter]

28 Sep 2021

PONE-D-20-37466R1 

Meta-Analysis illustrates possible role of lipopolysaccharide
(LPS)-induced tissue injury in nasopharyngeal carcinoma (NPC) pathogenesis 

Dear Dr. Allen:

I'm pleased to inform you that your manuscript has been deemed suitable for publication in PLOS ONE. Congratulations! Your manuscript is now with our production department. 

Kind regards, 

on behalf of

Dr. Srinivas Mummidi 

Academic Editor

PLOS ONE